# Tarantula toxins use common surfaces for interacting with Kv and ASIC ion channels

**Kanchan Gupta[1,2†], Maryam Zamanian[1,2†], Chanhyung Bae[1,2,4†], Mirela Milescu[1,2,5], Dmitriy Krepkiy[1,2], Drew C Tilley[3], Jon T Sack[3], Vladimir Yarov-Yarovoy[3], Jae Il Kim[4], Kenton J Swartz[1,2]\***

[1]Molecular Physiology and Biophysics Section, Porter Neuroscience Research Center, National Institutes of Health, Bethesda, United States; [2]National Institute of Neurological Disorders and Stroke, National Institutes of Health, Bethesda, United States; [3]Department of Physiology and Membrane Biology, University of California, Davis, Davis, United States; [4]Department of Life Science, Gwangju Institute of Science and Technology, Gwangju, Republic of Korea; [5]Biology Division, University of Missouri, Columbia, United States

**Abstract** Tarantula toxins that bind to voltage-sensing domains of voltage-activated ion channels are thought to partition into the membrane and bind to the channel within the bilayer. While no structures of a voltage-sensor toxin bound to a channel have been solved, a structural homolog, psalmotoxin (PcTx1), was recently crystalized in complex with the extracellular domain of an acid sensing ion channel (ASIC). In the present study we use spectroscopic, biophysical and computational approaches to compare membrane interaction properties and channel binding surfaces of PcTx1 with the voltage-sensor toxin guangxitoxin (GxTx-1E). Our results show that both types of tarantula toxins interact with membranes, but that voltage-sensor toxins partition deeper into the bilayer. In addition, our results suggest that tarantula toxins have evolved a similar concave surface for clamping onto α-helices that is effective in aqueous or lipidic physical environments.

*For correspondence: swartzk@ ninds.nih.gov

†These authors contributed equally to this work

Competing interests: The authors declare that no competing interests exist.

## Introduction

Protein toxins from the venom of poisonous organisms target a wide variety of ion channel proteins, and they have been powerful tools for studying the physiology and molecular mechanisms of these crucial signaling proteins. For example, scorpion toxins that physically block the pore of K+ channels (*MacKinnon and Miller, 1988*) were initially used to identify pore-forming regions of the channel (*MacKinnon and Miller, 1989*), and to determine the stoichiometry of subunits (*MacKinnon, 1991*), inactivation gates (*MacKinnon et al., 1993*), as well as accessory subunits (*Morin and Kobertz, 2008*). Protein toxins can also interact with ion channels and allosterically modify the process by which the channel opens in response to the stimulus that activates the channel. For example, both tarantula and scorpion toxins can bind to voltage-sensing domains in voltage-activated ion channels and either retard or facilitate activation in response to changes in voltage (*Rogers et al., 1996*; *Swartz and MacKinnon, 1997a*, *1997b*; *Cestèle et al., 1998*; *Milescu et al., 2013*). This second class of toxins have been valuable tools to explore the mechanism of voltage sensing (*Phillips et al., 2005*; *Alabi et al., 2007*; *Campos et al., 2007*, *2008*) and the roles of individual voltage sensors in the mechanisms of activation and inactivation (*Campos et al., 2007*; *Bosmans et al., 2008*; *Campos et al., 2008*). Tarantula toxins have also been discovered that activate transient receptor potential (TRP) channels (*Bohlen et al., 2010*), and both tarantula toxin and snake toxins have been identified that activate acid-sensing ion channels (ASIC) (*Chen et al., 2006*; *Salinas et al., 2006*; *Bohlen et al., 2011*). For both the TRPV1 channel and ASICs, these toxins were used to stabilize specific

**eLife digest** Venomous animals like tarantulas or scorpions inject their prey with toxins to disable them. Some of these toxins work by altering the activity of proteins called ion channels, which are found within membranes in cells. These channels can allow potassium ions and/or other ions to pass through the membrane and have many important roles. For example, ion channels are involved in heart muscle contraction and allow information to travel between brain cells.

Researchers have used some of the toxins as tools to study how ion channel proteins operate. For example, a toxin produced by tarantulas called psalmotoxin binds to a type of ion channel called the acid-sensing ion channels (ASIC). Researchers have previously been able to visualize the three-dimensional structure of psalmotoxin attached to ASIC, which revealed how the toxin binds to and alters the activity of the ion channel.

Another tarantula toxin called guangxitoxin is very similar to psalmotoxin, but it binds to a different type of potassium ion channel. It is thought that guangxitoxin binds to a site on these ion channels that is deep within the membrane, but it is not clear how this works. Here, Gupta, Zamanian, Bae et al. compare some of the structural and chemical properties of the two toxins.

The experiments show that both toxins interact with the membrane to enable them to bind with their target ion channels. However, guangxitoxin moves deeper into the interior of the membrane. Also, Gupta, Zamanian, Bae et al.'s findings suggest that both toxins use a similar surface that curves inwards to clamp onto their target ion channels. This common structural feature will be useful for designing experiments to visualize the three-dimensional structure of guangxitoxin bound to a potassium ion channel.

conformations for structure determination by cryo-electron microscopy and X-ray diffraction (*Baconguis and Gouaux, 2012*; *Dawson et al., 2012*; *Baconguis et al., 2014*).

Tarantula toxins such as hanatoxin and GxTx-1E are thought to target voltage-activated potassium (Kv) channels by partitioning into the lipid membrane and binding to their voltage-sensing domains to modify how the channel opens in response to changes in membrane voltage (*Swartz and MacKinnon, 1997a*, *1997b*; *Li-Smerin and Swartz, 2000*, *2001*; *Lee and MacKinnon, 2004*; *Wang et al., 2004*; *Phillips et al., 2005*; *Alabi et al., 2007*; *Milescu et al., 2007*, *2009*, *2013*; *Schmidt and MacKinnon, 2008*; *Bosmans et al., 2011a*; *Tilley et al., 2014*) (*Figure 1A*, right). The strong interaction of these tarantula toxins with lipid membranes is mediated by a prominent amphipathic surface on the toxins (*Figure 1F*) (*Takahashi et al., 2000*; *Lee et al., 2004*, *2010*), which facilitates their entry into the membrane (*Lee and MacKinnon, 2004*; *Milescu et al., 2007*, *2009*; *Jung et al., 2010*; *Mihailescu et al., 2014*). Although many aspects of this inhibitory mechanism have been well-studied (*Swartz and MacKinnon, 1997a*, *1997b*; *Li-Smerin and Swartz, 1998*, *2000*; *Lee et al., 2003*; *Wang et al., 2004*; *Phillips et al., 2005*; *Alabi et al., 2007*; *Milescu et al., 2007*, *2009*; *Bosmans et al., 2011a*; *Milescu et al., 2013*; *Mihailescu et al., 2014*; *Tilley et al., 2014*), no structures of complexes between tarantula toxins and Kv channels have been solved, limiting our understanding of how tarantula toxins dock onto voltage sensors and influence their conformational dynamics. However, a structurally related tarantula toxin named PcTx1 was recently crystalized in complex with ASIC1a (*Baconguis and Gouaux, 2012*; *Dawson et al., 2012*) (*Figure 1A*, left), revealing that the toxin clamps onto the thumb helix-5 (*Figure 1G*) and inserts an Arginine finger into the subunit interface to interact with residues involved in proton activation of the channel. PcTx1 modulates the gating properties of ASIC1 channels by enhancing the apparent affinity of these channels for protons, thereby promoting open and desensitized states of the channel (*Escoubas et al., 2000*, *2003*; *Chen et al., 2005*; *Salinas et al., 2006*; *Baconguis and Gouaux, 2012*; *Dawson et al., 2012*), an allosteric mechanism related to that employed by tarantula toxins targeting Kv channels. However, in contrast to the present model for tarantula toxins binding to Kv channels within the membrane (*Figure 1A*, right), the structures of PcTx1 bound to ASIC show that the toxin binds to the protein in aqueous solution far from the transmembrane domains of the channel (*Figure 1A* left). Interestingly, the overall fold of PcTx1 and voltage sensor toxins are quite similar (*Figure 1H*) and PcTx1 has significant amphipathic character (*Figure 1D*), similar to that seen in the structure of voltage-sensor toxins (*Figure 1E*).

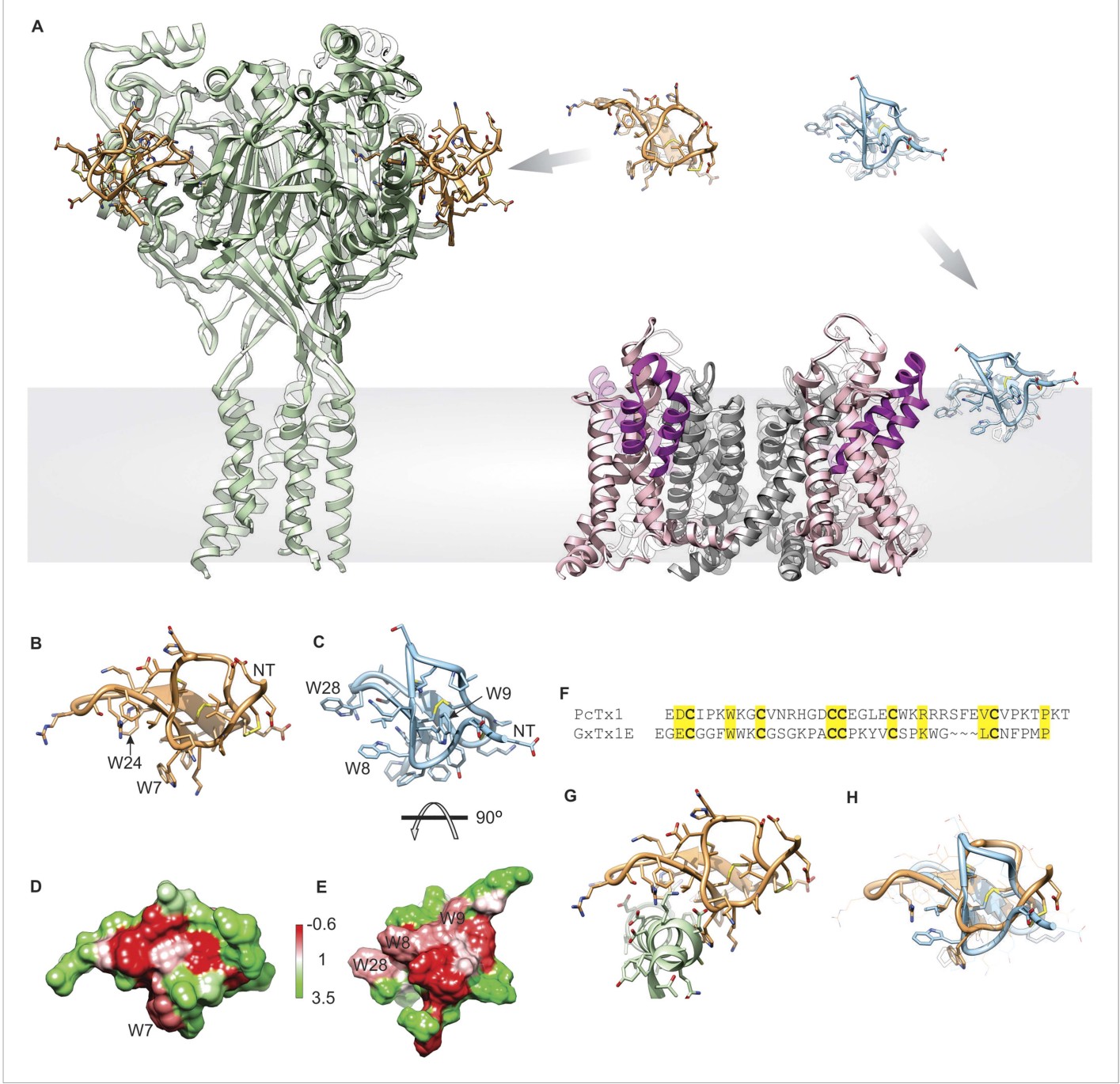

**Figure 1**. Structural comparison between tarantula toxins targeting ASIC and Kv channels. (**A**) Crystal structure of PcTx1 bound to the ASIC1a channel (left, PDB 4FZ0) and of the Kv1.2-Kv2.1 paddle chimera channel (right, PDB 2R9R) viewed from within the membrane. Voltage-sensor paddles and pore domain of the Kv1.2-Kv2.1 paddle chimera was colored purple and gray, respectively. (**B**, **C**) NMR structures of PcTx1 (**B**, PDB 2KNI) and GxTx-1E (**C**, PDB 2WH9) oriented by superimposing the toxin backbones. NT denotes N terminus. (**D**, **E**) Surface profiles of PcTx1 (**D**) and GxTx-1E (**E**) colored using the Hessa-Heijne hydrophobicity scale in kcal mol⁻¹ (*Hessa et al., 2005*). The orientation of PcTx1 is identical to those shown in **B**, **G** and **H**, whereas the structure of GxTx-1E was rotated 90° about the indicated axis to better illustrate its amphipathic character. (**F**) Sequence alignment of PcTx1 and GxTx-1E. Conserved residues are highlighted yellow. (**G**) Structure of PcTx1 bound to helix-5 of ASIC1a (PDB 4FZ0). (**H**) NMR Structures of PcTx1 and GxTx-1E superimposed based on their backbone folds. Side chains of conserved residues are shown as sticks.

The structural similarities of tarantula toxins targeting Kv and ASIC channels, when considered in the context of the unrelated structures of Kv and ASIC channels, as well as the distinct environments in which these toxins bind to their target ion channels, prompted us to undertake comparative studies of the membrane interaction properties and channel binding surfaces of the two classes of tarantula toxins. Does the amphipathic surface of PcTx1 enable the toxin to interact with membranes even though it binds to ASIC in solution? If so, how does the interaction of PcTx1 with membranes compare with that observed for toxins targeting voltage sensors? Do the two classes of structurally related toxins use distinct surfaces for binding to Kv channels within the membrane and to ASIC channels in solution? Our results demonstrate that both classes of tarantula toxins interact with membranes, but that PcTx1 interacts more superficially when compare to toxins targeting voltage sensors. In addition, our results suggest that PcTx1 and the voltage-sensor toxins use related surfaces for binding to helices within their target channels even though these events occur in very different physical environments.

## Results

In the present study we focused our investigations on PcTx1, a tarantula toxin for which X-ray structures are available in complex with ASIC1a (*Baconguis and Gouaux, 2012*; *Dawson et al., 2012*), and GxTx-1E, a tarantula toxin related to hanatoxin (*Swartz and MacKinnon, 1995*, *1997a*, *1997b*; *Takahashi et al., 2000*; *Phillips et al., 2005*; *Alabi et al., 2007*; *Bosmans et al., 2008*; *Milescu et al., 2013*; *Tilley et al., 2014*) that interacts with the S3b-S4 paddle motif within the S1–S4 voltage-sensing domain of a variety of Kv channels (*Herrington et al., 2006*; *Milescu et al., 2009*; *Schmalhofer et al., 2009*; *Lee et al., 2010*; *Milescu et al., 2013*). GxTx-1E inhibits opening of the Kv2.1 and engineered Shaker Kv channels with relatively high affinity (*Herrington et al., 2006*; *Milescu et al., 2009*; *Schmalhofer et al., 2009*; *Lee et al., 2010*; *Milescu et al., 2013*; *Tilley et al., 2014*), it can be produced using solid-phase peptide synthesis, efficiently folded in vitro (*Lee et al., 2010*; *Tilley et al., 2014*) (see 'Materials and methods') and its solution structure has been solved (*Lee et al., 2010*). Both toxins have a similar spacing of six Cys residues that participate in forming disulfide bonds (*Figure 1B*), and NMR solution structures reveal they adopt similar folds (*Figure 1C,D,H*), with backbone root mean square deviation (RMSD) over 20–22 core residues of 1.0–1.2 Å and over all residues of 2.9 Å. The two toxins have five other positions that are well-conserved, but otherwise their primary sequences vary considerably (*Figure 1F*). PcTx1 contains two solvent-exposed Trp residues, one in loop 1 and one in loop 4, whereas GxTx-1E contains three, two in loop 1 and one in loop 4. The amphipathic character of these toxins can be seen in surface renderings of the toxin structures when coloring residues using the Hessa-Heijne hydrophobicity scale (*Hessa et al., 2005*), although it is interesting that the clusters of solvent-exposed hydrophobic residues are on distinct surfaces of the two toxins (*Figure 1D,E*).

## Interactions of tarantula toxins with lipid membranes

Previous studies have established that tarantula toxins targeting ion channel voltage sensors interact favorably with lipid membranes (*Lee and MacKinnon, 2004*; *Suchyna et al., 2004*; *Jung et al., 2005*; *Phillips et al., 2005*; *Milescu et al., 2007*, *2009*; *Swartz, 2007*; *Jung et al., 2010*; *Mihailescu et al., 2014*). We conducted experiments to assess whether PcTx1 partitions into membranes similar to voltage sensor toxins. One common approach to measuring membrane partitioning is Trp fluorescence, where the emission spectra of Trp residues on the toxins are shifted to lower wavelengths (blue shifted) when the toxin moves from an aqueous environment to a more constraining and hydrophobic environment of the lipid membrane (*Ladokhin et al., 2000*). Pronounced blue shifts in Trp fluorescence are observed with GxTx-1E when lipid vesicles containing a 1:1 mix of POPC and POPG are added to aqueous solutions of the toxin (*Figure 2A*), consistent with a previous study (*Milescu et al., 2009*). Plotting of fluorescence intensity of GxTx-1E in the blue shifted region (320 nm) of the emission spectra as a function of lipid concentration and fitting of a partition function (*Figure 2C*) yields a mole fraction $K_x$ of $4.6 \pm 0.8 \times 10^6$, indicating favorable interactions of the toxin with membranes. To investigate whether PcTx1 can interact with membranes, we performed similar experiments and observed detectable blue shifts in the emission spectra of the toxin upon the addition of lipid vesicles (*Figure 2B*). The blue shifts in the case of PcTx1 were very small (maximal shift of ~ 2 nm), however, they varied as a function of lipid concentration and saturated at high lipid concentrations, suggestive of an interaction between PcTx1 and lipid membranes.

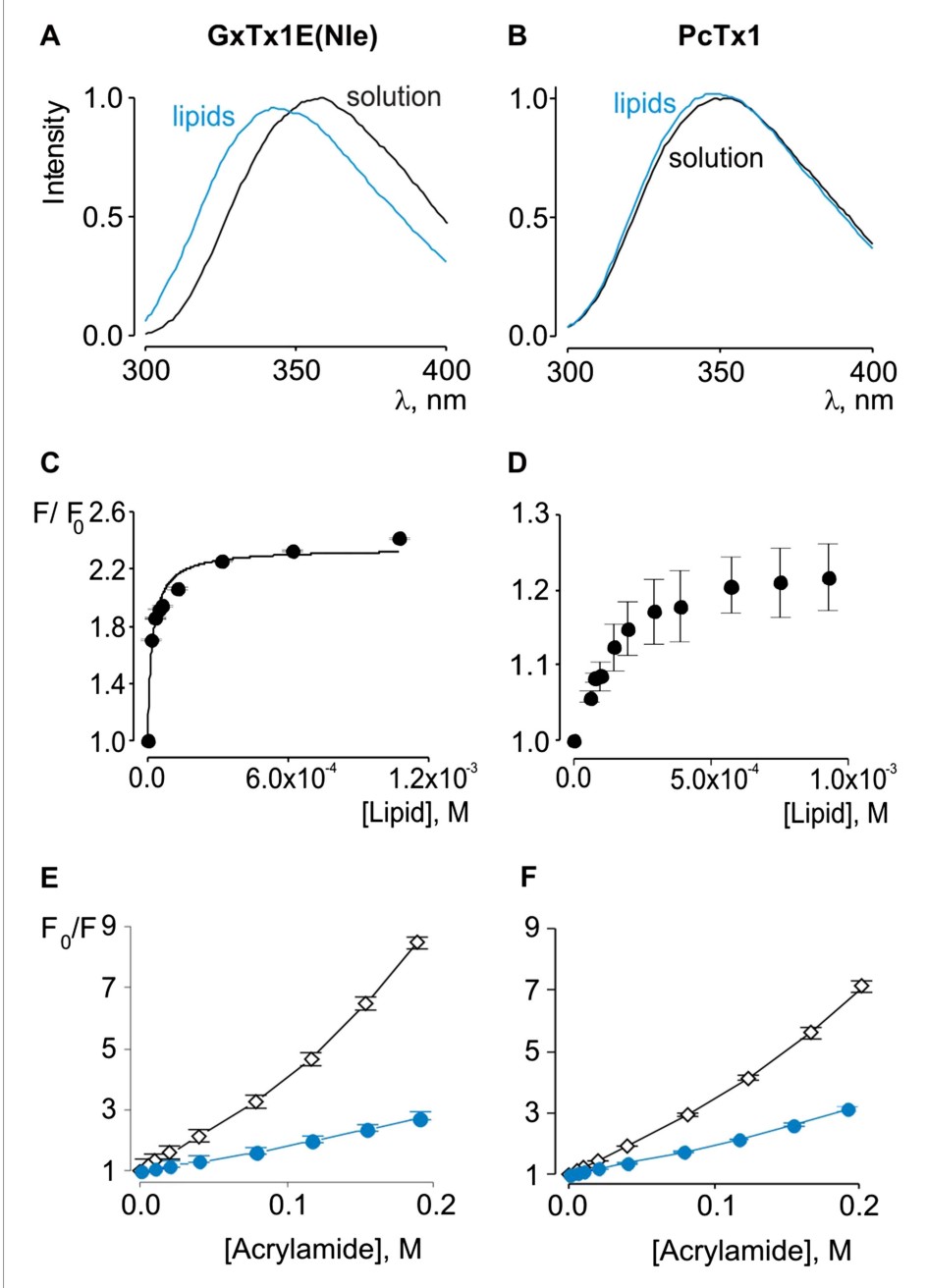

**Figure 2**. Interaction of GxTx-1E(Nle) and PcTx1 with lipid vesicles detected with intrinsic Trp fluorescence. (**A**, **B**) Fluorescence emission spectra of GxTx-1E(Nle) and PcTx1 in the absence (black) or presence of lipid vesicles composed of a 1:1 mix of POPC:POPG (blue). Lipid concentration was 1.0 mM. (**C**, **D**) Fluorescence intensity at 320 nm plotted as a function of available lipid concentration for GxTx-1E(Nle) or PcTx1. Smooth curve corresponds to a partition function with $K_x = (4.6 \pm 0.8) \times 10^6$ and $F/F_0^{max} = 2.3 \pm 0.05$ for GxTx-1E(Nle). (**E**, **F**) Stern-Volmer plots for acrylamide quenching of GxTx-1E(Nle) and PcTx1 in solution (~2 μM, black diamonds) and in the presence of lipid vesicles (1.0 mM, blue circles). In all cases data points are the mean ± SEM ($n = 3$).

Given the small blue shift observed for PcTx1, we further investigated its interactions with membranes by determining whether lipid vesicles could protect the toxin from quenching by acrylamide, a water soluble quencher of Trp fluorescence. In the case of GxTx-1E in aqueous solution, we observed that the Trp fluorescence of the toxin is quenched by the addition of acrylamide in

a concentration-dependent manner (*Figure 2E*), and fitting of the quenching relationship with a linear function yields a slope (Stern–Volmer constant; $K_{sv}$) of 37.7 $M^{-1}$. Addition of lipid vesicles decreased the $K_{sv}$ to 9.1 $M^{-1}$ (*Figure 2E*), indicating that membranes can shield the Trp residues in GxTx-1E from the soluble quencher. When these experiments were performed with PcTx1, we again observed strong quenching by acrylamide in aqueous solution ($K_{sv}$ of 29.2 $M^{-1}$) and robust protection of quenching by the addition of lipid vesicles ($K_{sv}$ of 11.2 $M^{-1}$; *Figure 2F*), confirming that PcTx1 can interact with membranes. Varying the concentration of lipid vesicles in the presence of a fixed concentration of acrylamide also resulted in robust and concentration-dependent protection of acrylamide quenching for both GxTx-1E (*Figure 3A,C*) and PcTx1 (*Figure 3B,D*). When the increase in maximal fluorescence intensity was plotted against the available lipid concentration and a partition function fit to the data, a $K_{dx}$ value of $(1.1 \pm 0.2) \times 10^7$ was obtained for GxTx-1E, in reasonable agreement with the mole fraction partition coefficient for the toxin determined using blue-shifts in Trp fluorescence. In the case of PcTx1, we obtained a value of $(4.3 \pm 0.6) \times 10^6$, demonstrating that PcTx1 interacts strongly with membranes.

The results thus far reveal that both classes of tarantula toxins interact quite favorably with lipid membranes. Although the structure of PcTx1 shows the presence of a clear hydrophobic cluster of residues on one face of the toxin, it is smaller than seen in structures of voltage-sensor toxins (*Figure 1D,E*). For example, the solvent accessible surface area for the hydrophobic surface of PcTx1 is 380 $Å^2$, which compares with a value of 703 $Å^2$ for GxTx-1E. In addition, the hydrophobic surfaces on the two toxins are situated on distinct surfaces (*Figure 1D,E*). These differences between PcTx1 and GxTx-1E raise the possibility that the two classes of toxin might exhibit distinct depths of penetration into membranes. To estimate the depth to which these toxins penetrate lipid membranes, we compared the extent of quenching of Trp fluorescence by bromine atoms attached at different positions along lipid acyl chains (*Ladokhin, 1997*, *1999*, *2014*). In previous studies, the analysis of quenching profiles for lipids brominated at three positions along the acyl chain of POPC (from most superficial to deepest; 6,7-diBr, 9,10-diBr or 11,12-diBr) have shown the strongest quenching for bromines attached near the middle of the lipid tail (9,10-diBr), and analysis of these profiles suggested that Trp residues in hanatoxin, SGTx (a close relative of hanatoxin) and VSTx are positioned 8–9 Å from the center of the bilayer (*Phillips et al., 2005*; *Swartz, 2007*; *Jung et al., 2010*; *Mihailescu et al., 2014*). When we performed quenching experiments with these three brominated lipids and GxTx-1E, we observed strong quenching that is similar to what has been reported previously for other voltage-sensor toxins (*Figure 3E*) (*Phillips et al., 2005*; *Swartz, 2007*; *Jung et al., 2010*; *Mihailescu et al., 2014*), confirming an intimate interaction of the toxin with the hydrophobic core of the membrane. However, we observed only relatively small differences between the three brominated lipids, precluding further quantitative analysis of the profiles to obtain a specific depth of penetration. We suspect that this unique pattern of quenching by brominated lipids for GxTx-1E might be caused by the presence of three Trp residues in the toxin, each of which will likely be in distinct positions when the toxin interacts with membranes (see below). Interestingly, when we undertook similar experiments with PcTx1, we observed only very weak quenching by brominated lipids, again with only relatively small differences between the three lipids (*Figure 3F*). The measurable quenching observed for PcTx1 confirms that this toxin interacts with membranes, however, the much weaker quenching when compared to voltage-sensor toxins indicates that this toxin interacts considerably more superficially with membranes.

## PcTx1 sensitivity is not readily transferable to voltage-sensor toxin binding sites

The X-ray structure of PcTx1 bound to ASIC shows that the toxin interacts primarily with residues on one face of thumb helix-5, with an interface comprised of both hydrophobic and polar residue interactions (*Figure 1G*) (*Baconguis and Gouaux, 2012*; *Dawson et al., 2012*). PcTx1 also contains an Arg finger motif composed of R26, R27 and R28 that is required for activation of ASIC through interactions with channel residues situated in a crevice between subunits. In the case of voltage-sensor toxins such as hanatoxin and GxTx-1E, the S3b helix within the voltage-sensing domains of Kv channels contains the most influential determinants of toxin binding, and these also involve a combination of hydrophobic and polar residue interactions (*Swartz and MacKinnon, 1997b*; *Li-Smerin and Swartz, 2000*, *2001*; *Alabi et al., 2007*; *Milescu et al., 2009*). Given the common helical targets of PcTx1 and voltage-sensor toxins, and the finding that PcTx1 can interact with

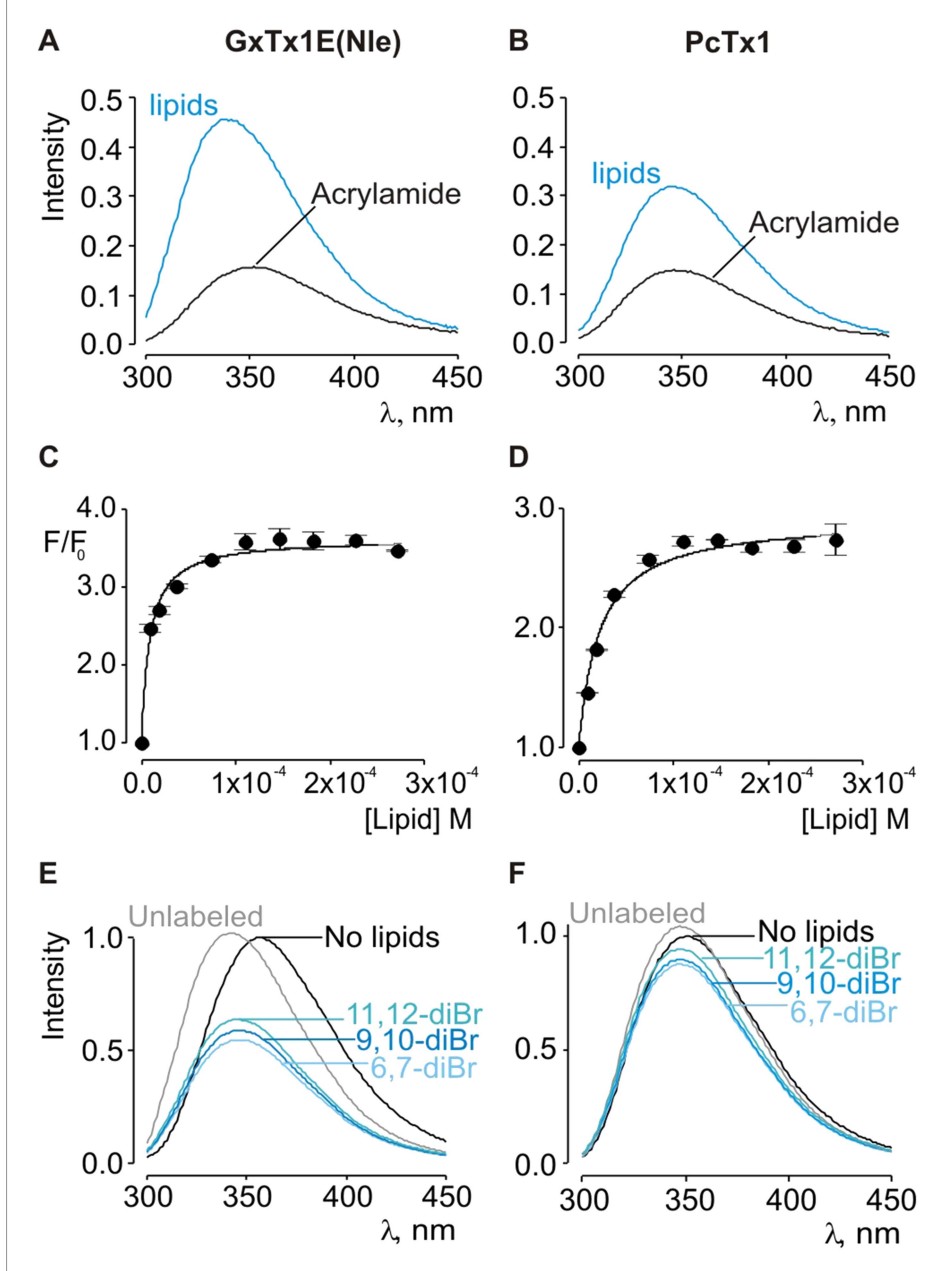

**Figure 3**. Interaction of GxTx-1E(Nle) and PcTx1 with lipid vesicles using acrylamide dequenching and quenching with brominated lipids. (**A**, **B**) Fluorescence emission spectra of GxTx-1E(Nle) and PcTx1 in the absence (black) or presence of lipid vesicles composed of 1:1 mix of POPC:POPG (blue) in a solution containing 0.2 M acrylamide. Lipid concentration was 1.0 mM. Fluorescence intensity was normalized to that measured for the toxin in control solution in the absence of quencher. (**C**, **D**) Maximum fluorescence intensity plotted as a function of available lipid concentration for GxTx-1E(Nle) and PcTx1. Smooth curves correspond to dequenching functions with $K_{dx} = (1.1 \pm 0.2) \times 10^7$ and $F/F_0^{max} = 3.5 \pm 0.02$ for GxTx-1E(Nle) and $K_{dx} = (4.3 \pm 0.6) \times 10^6$ and $F/F_0^{max} = 2.7 \pm 0.08$ for PcTx1 (**D**). (**E**, **F**) Depth-dependent quenching of tryptophan fluorescence by brominated (diBr) phosphatidylcholines labeled at different positions on the acyl chain of POPC. Fluorescence emission spectra of GxTx1E(Nle) and PcTx1 in the absence (black) or presence of vesicles comprised of unlabeled (gray) or brominated lipids (blue) present at a concentration of 1.2 mM. Vesicles comprised of either unlabeled or brominated lipids contained a 1:1 mix of POPC:POPG, and the brominated lipid was POPC. In all cases data points are the mean ± SEM (n = 3).

membranes, we wondered whether it might be possible to transfer sensitivity to PcTx1 into a Kv channel by exchanging the S3b helix with helix-5 from ASIC. We did not attempt to transfer channel residues interacting with the arginine finger motif because these residues and helix-5 are noncontiguous in the ASIC sequence. PcTx1 has nM affinity for ASIC1a (*Chen et al., 2006*; *Dawson et al., 2012*), but even at very high concentrations (5 µM) has no effect on the Kv2.1 channel (*Figure 4*, *Figure 4—figure supplement 1B*). Previous chimera studies have shown that the voltage-sensor paddle, a helix-turn-helix motif comprised of S3b and S4 helices, can be transplanted between different channels that contain S1–S4 domains without disrupting voltage sensor function (*Alabi et al., 2007*; *Bosmans et al., 2008*, *2011b*; *Kalia and Swartz, 2013*), making us optimistic that helix-5 transfer might be possible.

We chose to make chimeras between ASIC1a and Kv2.1 by transferring helix-5 of ASIC1a into the Kv2.1, a channel that is inhibited by hanatoxin and GxTx-1E (*Swartz and MacKinnon, 1997a*; *Lee et al., 2003*, *2010*; *Phillips et al., 2005*; *Milescu et al., 2009*; *Tilley et al., 2014*) and has been a successful recipient of transplanted peptide toxin binding sites (*Alabi et al., 2007*; *Bosmans et al., 2008*, *2011b*; *Kalia and Swartz, 2013*). We generated a total of five chimeras where nine residues of helix-5 replaced nine residues in the S3b helix in five different frames starting at the N-terminal end of the S3b helix (*Figure 4—figure supplement 1A*). All five chimeras gave rise to robust voltage-activated K$^+$ currents and displayed voltage-activation relations that were shifted to more depolarized voltages when compared to the wild-type Kv2.1 channel (*Figure 4—figure supplement 1C–G*), consistent with disruption of the interactions between S3b and S4 helices that have been shown to stabilize voltage sensors in an activated state (*Xu et al., 2013*). Although these relatively radical chimeras were all functional, extracellular application of high concentrations of PcTx1 had no discernible effect on the activity of any of them (*Figure 4—figure supplement 1C–G*). We also constructed models of PcTx1 bound to the five chimeras by transplanting PcTx1 bound to helix-5 from the complex structure (*Baconguis and Gouaux, 2012*) into the X-ray structure of the Kv1.2/2.1 paddle chimera (*Long et al., 2007*) (*Figure 4—figure supplement 1H–Q*). As expected, several of the models predict steric clashes between PcTx1 and neighboring helices within the S1–S4 domain, whereas in others the toxin could be readily accommodated when bound to helix-5. Even in this latter category of models, however, PcTx1 would need to partition relatively deeply into the membrane in order to bind to helix-5 in the chimeras, a possibility that seems unlikely given the relatively superficial interaction of the toxin we inferred with brominated lipids.

## GxTx-1E and PcTx1 use overlapping surfaces for binding to helices

To further compare how PcTx1 and GxTx-1E interact with helices in ASIC and Kv channels, we alanine-scanned GxTx-1E to identify residues that are most critical for interaction with the Kv2.1 channel. We generated Ala mutants for 29 out of 36 residues of GxTx-1E (excluding one native Ala and 6 Cys residues) using solid phase chemical synthesis, and successfully folded all but one in mg quantities (see 'Materials and methods'). To determine the apparent affinity of toxin mutants for the resting state of Kv2.1, we obtained conductance (G)-voltage (V) relationships from negative holding voltages in the absence and presence of different concentrations of each mutant (*Figure 4A,B*) and estimated fractional occupancy of the channel from the fractional inhibition for relatively weak depolarizations (−20 to 0 mV) where toxin bound channels do not open, as previously described (*Swartz and MacKinnon, 1997a*, *1997b*; *Li-Smerin and Swartz, 2000*; *Phillips et al., 2005*; *Milescu et al., 2009*). We obtained estimates of the apparent affinity for each mutant, and observed a wide range of different phenotypes, from small changes in affinity to almost 400-fold weakening of toxin affinity for Kv2.1 (*Figure 4D*; *Table 1*). When perturbation energies (∆∆G values) for each mutant were mapped onto the NMR structure of GxTx-1E, a well-defined surface of the toxin was identified where mutations having dramatic effects on the apparent affinity of the toxin cluster together (*Figure 5A*). This surface contains many aromatic residues (F7, W8, W9, Y22 and W28), one aliphatic hydrophobic residue (L30) and several polar residues (K10, K15, S25, K27 and N32). The mutations producing the most dramatic perturbations on this surface were for mutations of aromatic residues, whereas those for the polar residues were relatively modest (*Figure 5A*; *Table 1*).

Because voltage-sensor toxins bind to Kv channels within the membrane, it is possible that some of the mutations we have identified influence the apparent affinity of the toxin by weakening the interaction of the toxin with membranes rather than exclusively weakening the protein–protein interaction. To verify that the surface we identified is involved in binding to Kv2.1, we investigated

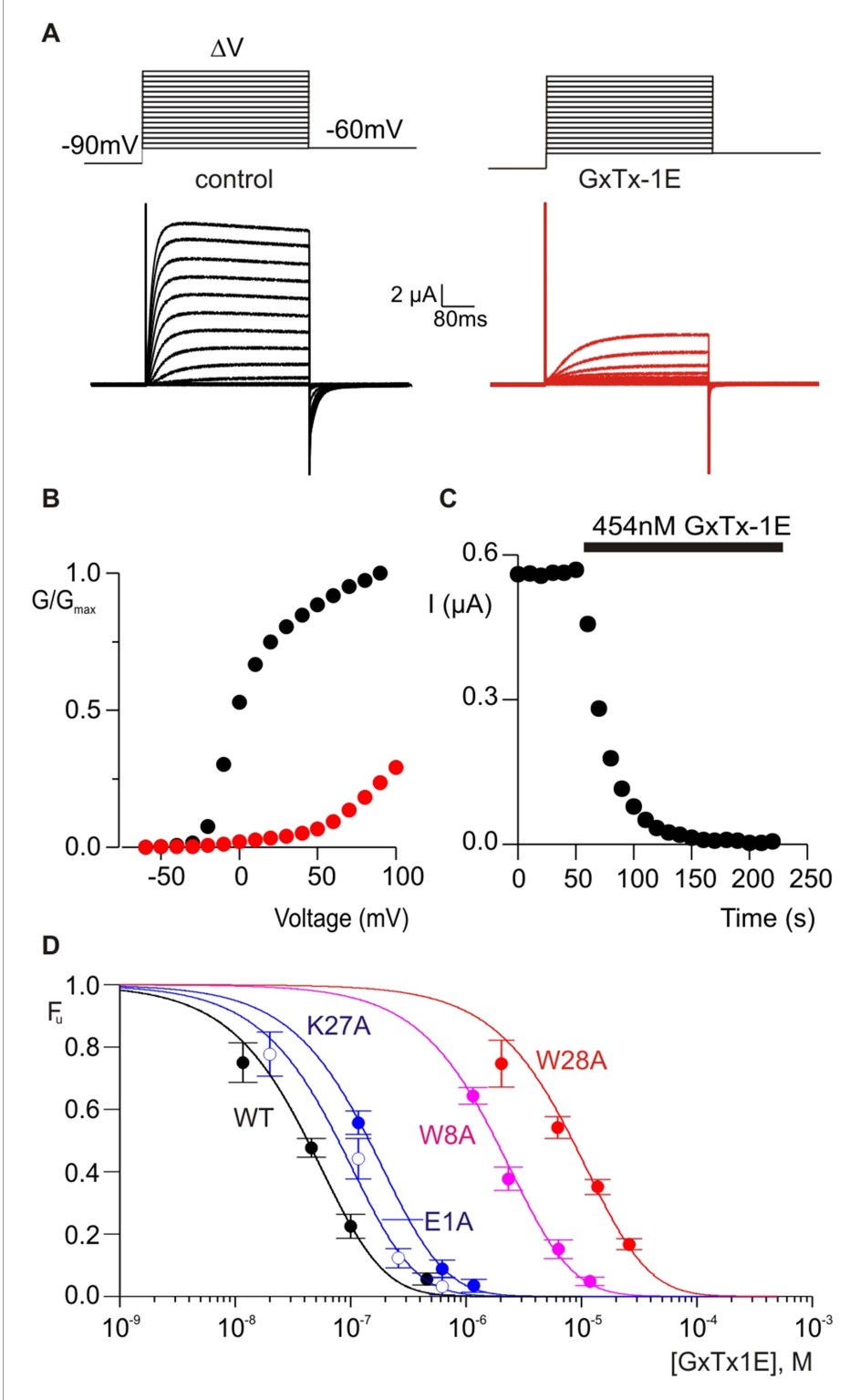

**Figure 4**. Determination of apparent affinity for mutants of GxTx-1E. (**A**) Families of macroscopic ionic currents elicited by test depolarizations before (black) and after (red) addition of 454 nM GxTx-1E(Nle) for an oocyte expressing the Kv2.1 channel. (**B**) G-V relations obtained from tail current amplitudes before and after addition of toxin. (**C**) Time course of inhibition of the Kv2.1 channel by 454 nM GxTx-1E(Nle). Steady-state currents were measured at the end of test depolarizations to −10 mV, elicited every 10 s. Holding voltage was −100 mV. (**D**) Concentration-dependence for inhibition of the Kv2.1 channel by GxTx-1E(Nle) and representative toxin

*Figure 4. continued on next page*

*Figure 4. Continued*

mutants. Fraction unbound ($F_u$) was measured using steady state current values before and after addition of toxin from time course of toxin inhibition as illustrated in C using weak depolarizations (−20 mV to +10 mV). Data points are mean ± SEM for 3 to 13 cells at each concentration. E1A and K10A are examples of mutants that produce weak perturbations in $K_d$ ($\Delta\Delta G$ < 1 kcal mol$^{-1}$), while W8A and W28A are examples of mutants that have larger perturbations ($\Delta\Delta G$ > 2 kcal mol$^{-1}$). See *Table 1* for all $K_d$ and $\Delta\Delta G$ values.

The following figure supplement is available for figure 4:

**Figure supplement 1**. Influence of PcTx1 on chimeras between Kv2.1 and ASIC1a.

membrane interactions for each of the three Trp residues (W8, W9 and W28), positions where mutations have some of the largest perturbation energies. Even though the apparent affinities of these mutants for Kv2.1 ranged from 26 to almost 200-fold lower (*Figure 4D*; *Table 1*), we obtained

**Table 1**. Affinities and perturbation energies for mutants of GxTx-1E

| Toxin | $K_d$ (nM) | $K_d^{mut}/K_d^{wt}$ | $\Delta\Delta G$ (kcal mol$^{-1}$) |
|---|---|---|---|
| Gxtx-1E(Nle) | 224 ± 25 | 1.0 | – |
| E1A | 414 ± 38 | 1.8 | 0.36 |
| G2A | 297 ± 50 | 1.3 | 0.17 |
| E3A | 125 ± 62 | 0.6 | −0.35 |
| G5A | 13,652 ± 320 | 60.9 | 2.44 |
| G6A | 379 ± 51 | 1.7 | 0.31 |
| F7A | 89,033 ± 9413 | 397.4 | 3.55 |
| W8A | 9641 ± 504 | 43.0 | 2.23 |
| W9A | 5837 ± 1850 | 26.1 | 1.93 |
| K10A | 984 ± 90 | 4.4 | 0.88 |
| G12A | 21,755 ± 1086 | 97.1 | 2.71 |
| S13A | 545 ± 56 | 2.4 | 0.53 |
| G14A | 461 ± 89 | 2.1 | 0.43 |
| K15A | 823 ± 127 | 3.7 | 0.77 |
| P16A | 22,967 ± 2812 | 102.5 | 2.75 |
| P20A | 652 ± 73 | 2.9 | 0.63 |
| K21A | 440 ± 82 | 2.0 | 0.40 |
| Y22A | 35,337 ± 7464 | 157.7 | 3.00 |
| V23A | 516 ± 60 | 2.3 | 0.50 |
| S25A | 1169 ± 64 | 5.2 | 0.98 |
| P26A | 813 ± 99 | 3.6 | 0.76 |
| K27A | 763 ± 49 | 3.4 | 0.73 |
| W28A | 44,298 ± 2471 | 197.7 | 3.14 |
| L30A | 2474 ± 278 | 11.0 | 1.42 |
| N32A | 1228 ± 166 | 5.5 | 1.01 |
| F33A | 491 ± 18 | 2.2 | 0.47 |
| P34A | 348 ± 60 | 1.6 | 0.26 |
| Nle35A | 306 ± 19.8 | 1.4 | 0.19 |
| P36A | 250 ± 22 | 1.1 | 0.07 |

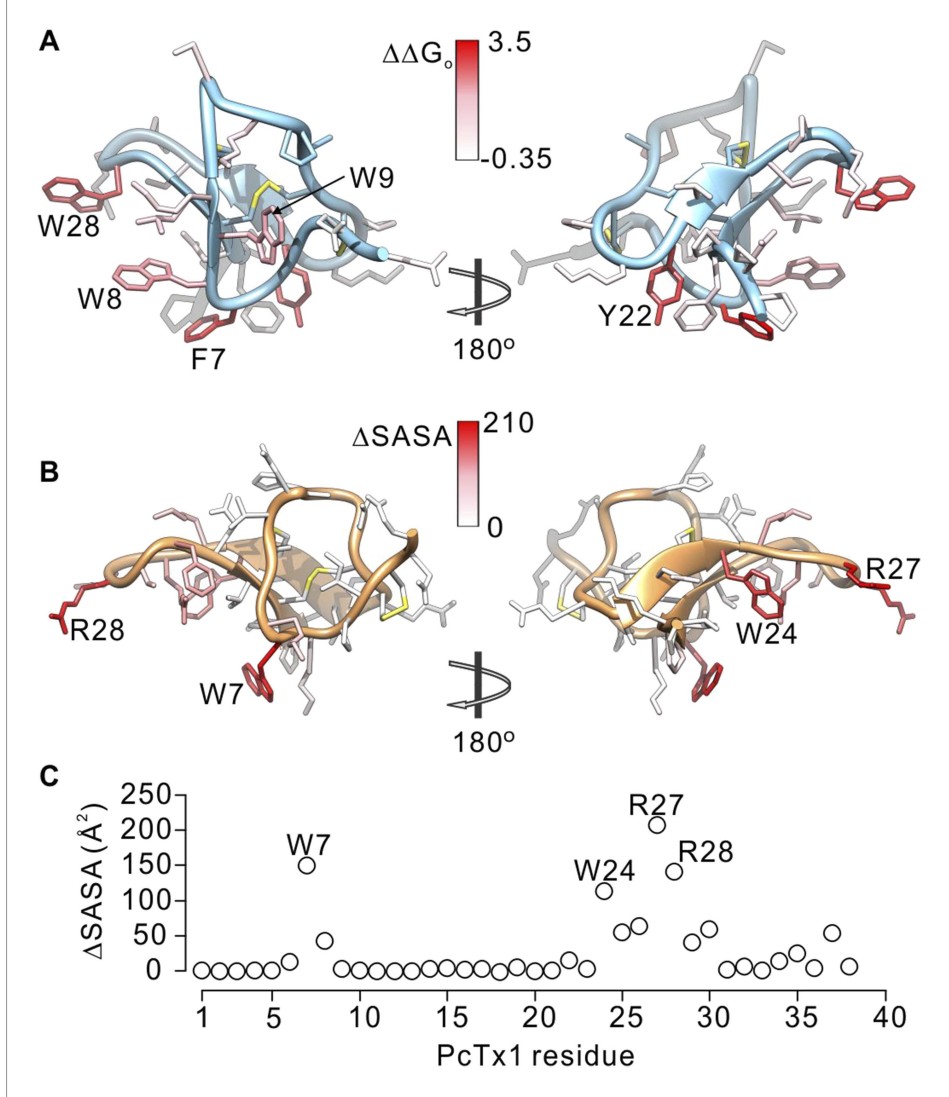

**Figure 5**. Comparison of residues on PcTx1 and GxTx-1E involved in receptor binding. (**A**) GxTx-1E residues colored by perturbation in apparent affinity ($\Delta\Delta G_o$ in kcal mol$^{-1}$). Several side-chains with the largest perturbation energies are labeled. P16 was not colored since its mutation to Ala likely perturbs the structure of the toxin. (**B**) PcTx1 residues colored by changes in solvent accessible surface area ($\Delta$SASA in Å$^2$) of the toxin upon binding to ASIC1a. SASA was calculated using UCSF-Chimera with a 1.4 Å sphere for the structure of PcTx1 alone or in complex with ASIC 1a (PDB 4FZ0). Residues with the largest $\Delta$SASA (W7, W24, R27 and R28) are labeled. (**C**) $\Delta$SASA plotted against PcTx1 residue number.

The following figure supplements are available for figure 5:

**Figure supplement 1**. Interaction of tryptophan mutants of GxTx-1E(Nle) with lipid vesicles detected with intrinsic Trp fluorescence.

**Figure supplement 2**. Depth-dependent quenching of tryptophan fluorescence by brominated (diBr) phosphatidylcholines.

**Figure supplement 3**. Comparison of the effects of GxTx-1E and SGTx1 mutation on apparent affinity and membrane partitioning.

$K_x$ values that were within a factor of two for the wild-type toxin (*Figure 5—figure supplement 1*). We also measured quenching of the three Trp mutants by brominated lipids and found that each exhibited robust quenching (*Figure 5—figure supplement 2*). It is notable that the blue-shifts observed with membrane partitioning vary for the three Trp mutants, as does the extent of quenching by brominated lipids (*Figure 5—figure supplement 2*), suggesting that the three Trp residues reside in non-equivalent environments within the membrane. Taken together, these collective data with GxTx-1E mutants strongly support the notion that the active surface identified by mutagenesis (*Figure 5A*) is involved in binding to the voltage-sensing domain. To compare this surface of GxTx-1E with the binding surface of PcTx1, we calculated the change in <u>s</u>olvent <u>a</u>ccessible <u>s</u>urface <u>a</u>rea (ΔSASA) of PcTx1 upon binding to ASIC1a, and compared it to the binding surface identified in GxTx-1E (*Figure 5A–C*). The comparison reveals an extensive overlap between the active surfaces of the two toxins responsible for binding to ASIC1a and Kv2.1 channels, indicating that the two classes of toxins use related surfaces for interacting with helices within structurally unrelated ion channel proteins.

## Discussion

The objective of this study was to compare membrane interactions and channel binding surfaces of the voltage-sensor toxin GxTx-1E and the ASIC toxin PcTx1. The recent X-ray structures of complexes between PcTx1 and ASIC1a indicates that the tarantula toxin binds to the channel in aqueous solution (*Baconguis and Gouaux, 2012*; *Dawson et al., 2012*), whereas a growing body of work supports that idea that voltage-sensor toxins bind to voltage sensors within the lipid bilayer (*Lee and MacKinnon, 2004*; *Wang et al., 2004*; *Phillips et al., 2005*; *Alabi et al., 2007*; *Milescu et al., 2007, 2009, 2013*; *Schmidt and MacKinnon, 2008*; *Bosmans et al., 2011a*). The results of our experiments demonstrate that both classes of toxins interact with membranes (*Figures 2, 3*), a finding that is consistent with the amphipathic character evident in the solution structures of these toxins (*Figure 1*). However, our results also show that the interaction of PcTx1 with membranes is different than what has been observed for voltage-sensor toxins. Fluorescence spectroscopy results show that Trp residues on PcTx1 exhibit only very small blue shifts when interacting with membranes (*Figure 2*), and that the Trp fluorescence of the toxin is only weakly quenched by bromine atoms on the acyl chains of lipids (*Figure 3*). In contrast, voltage-sensor toxins typically exhibit strong blue shifts in Trp fluorescence and this is robustly quenched by brominated lipids (*Figures 2, 3*). Although these properties have been consistently observed for other voltage-sensor toxins (*Jung et al., 2005*; *Phillips et al., 2005*; *Milescu et al., 2007, 2009*; *Swartz, 2007*; *Jung et al., 2010*; *Mihailescu et al., 2014*), the stark differences observed here for PcTx1 demonstrate that tarantula toxins of similar structure do not necessarily have similar membrane interacting properties. The membrane interactions of voltage-sensor toxins are unique in how far they can penetrate the lipid bilayer, further supporting the idea that this class of toxins interacts with voltage sensors within the lipid bilayer.

In addition to providing a valuable comparison with voltage-sensor toxins, the interaction between PcTx1 and membranes may be interesting in its own right. Although a complex structure of PcTx1 and ASIC shows that the PcTx1 binding site is located about 45 Å above from the lipid bilayer, superficial adsorption of the toxin to the membrane would be predicted to enhance the rate of complex formation due to a reduction in dimensionality of diffusion (*Axelrod and Wang, 1994*). Interestingly, membrane interactions have also been reported for charybdotoxin (*Ben-Tal et al., 1997*), a strongly cationic scorpion toxin that blocks the external pore of some potassium channels (*Miller, 1995*; *Banerjee et al., 2013*). The interaction of charybdotoxin with membranes is heavily influenced by electrostatic interactions, suggesting that the toxin interacts relatively superficially, similar to what we observed with PcTx1. The superficial nature of the interaction between either PcTx1 or charybdotoxin and lipid membranes could be important because it would allow these toxins to departition rapidly so that it can bind to its receptor above the membrane, or possibly to explore the protein surface as an extension of the membrane surface. The voltage-sensor toxins, in contrast, must bind to helices embedded within the membrane, and thus must venture deeper into the lipid bilayer.

The inhibitor cystine knot (ICK) fold is found in many toxins and is thought to provide a structurally stable scaffold for presentation of surfaces to bind to receptors and ion channel proteins (*Pallaghy et al., 1994*; *Norton and Pallaghy, 1998*), and in principle many unique surfaces on these toxins could be used for this purpose. However, our results show that GxTx-1E uses a surface to bind to the S3b helix within voltage sensors that is similar to that which PcTx1 uses to bind to helix-5 in ASIC

(*Figure 5*), even though the former binds within a membrane environment and the latter in aqueous solution. The underlying rationale for using common surfaces is likely related to the fact that both GxTx-1E and PcTx1 bind directly to α-helices. The X-ray structure of PcTx1 in complex with ASIC shows that the toxin clamps onto a solvent-exposed lateral surface of helix-5 (*Figure 1*) (*Baconguis and Gouaux, 2012*; *Dawson et al., 2012*), using a somewhat concave surface of the toxin formed by residues in loop 1 and loop 3, including both hydrophobic (W7, W24, F30, and P35) and polar residues (K8, K25, R26, S29 and T37). The equivalent surface of GxTx-1E is also formed by residues in loop 1 and loop 3 (*Figure 5*), which contain a similar variety of hydrophobic (F7, W8, W9, Y22, W28 and L30) and polar residues (K10, K15, S25, K27 and N32) where mutations weaken the apparent affinity of the toxin. This surface is also similar to that identified on SGTx1 (*Wang et al., 2004*; *Milescu et al., 2007*), a tarantula toxin related to hanatoxin and GxTx-1E that inhibits the Kv2.1 channel. Although the apparent affinity of the most critical mutants on SGTx1 could not be accurately determined because the affinity of the toxin is quite low (*Wang et al., 2004*), residues having disproportionately large effects on the apparent affinity of the toxin compared to the strength of membrane partitioning cluster together on a surface that is similar to what we identified here for GxTx-1E (*Figure 5—figure supplement 3*) (*Milescu et al., 2007*). Given the similar binding faces for α-helices on 3 peptide toxins that share little commonality besides an tri-cystine scaffold, we suggest that this face of the ICK fold forms a shape complement (*Jones, 2012*) for α-helices that may be exploited by many other peptide ligands.

To further explore the structure of GxTx-1E bound to voltage-sensors, we constructed models of GxTx-1E bound to the X-ray structure of the activated/open Kv1.2/Kv2.1 paddle chimera by superimposing GxTx-1E with PcTx1 bound to ASIC and then transposing the toxin-helix interactions into the paddle chimera. We constructed five models where the toxin-helix interaction occurs in different registers in the S3b helix, similar to what we did in making the helix-5/paddle chimera models, and identified three models where the toxin could be accommodated without steric clashes with the channel (*Figure 6*, *Figure 6—figure supplements 1, 2*). Although all three models position critical residues on GxTx-1E where they can interact with the S3b helix, our preferred model (*Figure 6*; *Figure 6—figure supplements 1E, 2E*) positions the largest number of critical residues in GxTx-1E where they can interact with residues in the S3b helix. In addition, in this model the toxin interacts with many of the most critical residues within the S3b helix that have been identified using mutagenesis (*Figure 6*) (*Swartz and MacKinnon, 1997b*; *Li-Smerin and Swartz, 2000, 2001*; *Milescu et al., 2009*), with a placement of the toxin within the membrane that is compatible with the depth of partitioning inferred using spectroscopic and neutron diffraction approaches (*Phillips et al., 2005*; *Jung et al., 2010*; *Mihailescu et al., 2014*). Voltage-sensor toxins like GxTx-1E bind tighter to and stabilize the resting/closed state of Kv channels, however, they can remain bound when the voltage sensors activate and the channel opens (*Phillips et al., 2005*; *Tilley et al., 2014*). Thus, although our preferred model of GxTx-1E bound to the activated/open Kv1.2/Kv2.1 paddle chimera represents the toxin bound to a lower affinity activated state of the Kv channel, it nicely illustrates how GxTx-1E could interact with the S3b helix in ways that closely resemble what is seen in the X-ray structure of PcTx1 bound to ASIC. In the future it will be interesting to refine models of the complex by attempting to form bridges between toxin and channel, and ultimately solving structures of these fascinating complexes.

## Materials and methods

### Solid phase peptide synthesis of tarantula toxins

Peptide synthesis was conducted on an Applied Biosystems model 433A peptide synthesizer as previously described (*Lee et al., 2010*). The linear precursor was synthesized using solid-phase methodology with Fmoc chemistry, starting from Fmoc-Pro-Wang resin (Fmoc-Thr-Wang resin for PcTx1) using a variety of blocking groups for the protection of the amino acids. All the GxTx1E peptides were synthesized with norleucine (Nle) in place of Met35 to avoid oxidation of the toxin. A 4 mol excess of Fmoc amino acid, DIC, and Cl-HOBt were used for amino acid activation. After trifluoroacetic acid cleavage, the crude linear peptide was extracted with 2 M acetic acid and diluted to a final concentration of 25 μM in a solution containing 0.1 M ammonium acetate, 1 M guanidine-HCl and 2.5 mM reduced/0.25 mM oxidized glutathione (pH 8.0 with aqueous $NH_4OH$) and stirred slowly at 4°C for 3 days. The folding reaction was monitored with RP-HPLC and the crude oxidized product was purified by preparative RP-HPLC with a C18 silica column. The purity of the synthetic PcTx1, GxTx1E and the Ala mutants was confirmed by analytical RP-HPLC and MALDI-TOF-MS

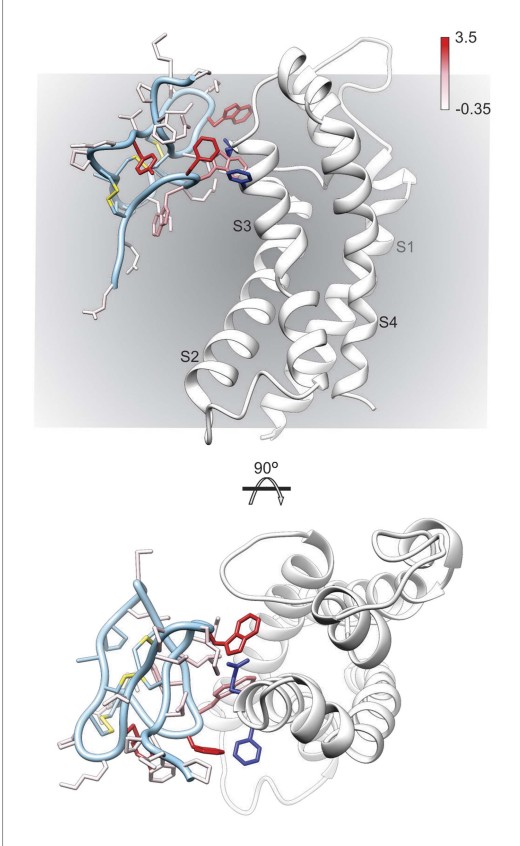

**Figure 6**. Model of GxTx-1E bound to the S3b helix of the Kv1.2/2.1 paddle chimera. The preferred model of GxTx-1E bound to the X-ray structure of the voltage-sensing domain of the Kv1.2/2.1 paddle chimera generated as described in 'Materials and methods'. The model maintains a relatively superficial position for the toxin relative to the membrane (see *Figure 6—figure supplements 1, 2*). The complex is viewed from a transmembrane perspective (top) and from the extracellular perspective (bottom). GxTx-1E side-chains are colored by free energy perturbation scale ($\Delta\Delta G_o$ in kcal mol$^{-1}$). F271 and E274 of the channel, where in Kv2.1, mutation to alanine decreases apparent $K_d$ values for GxTx-1E (*Milescu et al., 2009*), are represented by blue colored side-chains.

The following figure supplements are available for figure 6:

**Figure supplement 1**. Model structures of GxTx-1E bound to the voltage-sensing domain of a Kv channel.

**Figure supplement 2**. Effects of GxTx-1E and Kv2.1 mutations on the energetics of toxin-channel interactions.

measurements. The concentration of the toxin was determined by measuring absorbance at 280 nm using calculated extinction coefficients (*Gill and von Hippel, 1989*).

## Construction of Kv2.1-ASIC1a chimeras and electrophysiological recording

Chimeras were generated using sequential PCR with the Kv2.1Δ7 channel (*Li-Smerin and Swartz, 1998*) as template. Primers encoding the 9-amino acid sequence of helix-5 from ASIC1a were utilized for overlap PCR. The DNA sequence of all constructs and mutants was confirmed by automated DNA sequencing. Complementary RNA (cRNA) was synthesized using T7 polymerase (mMessage mMachine kit; Ambion) after linearizing the DNA with appropriate restriction enzymes.

Oocytes from *Xenopus laevis* were removed surgically and incubated with agitation for 1 hr in a solution containing (in mM) 82.5 NaCl, 2.5 KCl, 1 MgCl$_2$, 5 HEPES, pH 7.6 (with NaOH), and collagenase (2 mg/ml; Worthington Biochemical). Defolliculated oocytes were injected with cRNA encoding the Kv2.1 channel constructs and incubated at 17°C in a solution containing (in mM) 96 NaCl, 2 KCl, 1 MgCl$_2$, 1.8 CaCl$_2$, 5 HEPES, pH 7.6 with NaOH, and gentamicin (50 μg/ml, GIBCO-BRL), for 16–72 hr before electrophysiological recording. Oocyte membrane voltage was controlled using an OC-725C oocyte clamp (Warner Instruments). Data were filtered at 1 kHz and digitized at 10 kHz. Microelectrode resistances were 0.1–1 MΩ when filled with 3 M KCl. The external recording solution for all recordings with GxTx1E and its analogues contained 50 mM RbCl, 50 mM NaCl, 10 mM HEPES (increased to 20 mM at 10 μM and higher concentrations of F7A toxin mutant), 1 mM MgCl$_2$ and 0.3 mM CaCl$_2$ at pH 7.6 (with NaOH). All experiments were performed at room temperature (~22°C). Voltage-activations relationships in the absence and presence of different concentrations of the synthesized toxin were recorded. The inhibitory activity of the synthetic toxins were examined against the Kv2.1 channel by taking steady-state current measurements following weak depolarizations at 10-s intervals before, during and after addition of toxin to the recording chamber until the effect of the toxin on the channel reached equilibrium. At weak depolarizations, leak and endogenous current contributions to steady-state current levels were accounted for by using clampfit to subtract current before activation of the channels at those voltages. Experiments with PcTx1 were carried out with external solution containing 50 mM KCl.

We examined the toxin occupancy of closed or resting channels at negative holding voltages where open probability is low and we estimated the fraction of unbound channels ($F_u$) using depolarizations that are too weak to open toxin-bound channels, as previously described (*Swartz and MacKinnon, 1997a*, *1997b*; *Li-Smerin and Swartz, 2000*; *Lee et al., 2003*; *Phillips et al., 2005*; *Alabi et al., 2007*; *Bosmans et al., 2008*; *Milescu et al., 2009*). For all toxins, we calculated the ratio of steady-state currents ($I/I_0$) at weak depolarizations before addition ($I_0$) of different concentrations of toxin and after the toxin effects reached equilibrium ($I$). Voltage-activation relations before and after addition of toxin indicated that the values of $I/I_0$ at these weak depolarizations were in the plateau phase where toxin-bound channels do not open. The apparent $K_d$ for each toxin was calculated assuming four independent toxin-binding sites per channel, with single occupancy being sufficient to inhibit opening in response to weak depolarizations: $K_d = ((1/(1 - F_u^{1/4})) - 1)$ [Toxin].

## Fluorescence spectroscopy

Large unilamellar vesicles (LUVs) were prepared by drying the phospholipids from a chloroform solution under a nitrogen stream. The dried lipid film was rehydrated in buffer containing 10 mM HEPES, 1 mM EDTA, pH 7.6 (HEB). The resulting dispersions were extruded through 100 nm pore size polycarbonate filters (Millipore Corp.) till the solution became clear. All fluorescence measurements were performed in quartz cuvettes with 1 cm path length. Fluorescence spectra (averaging three spectra) were recorded between 300 and 450 nm (5 nm band pass, 0° polarizer) using an excitation wavelength of 280 nm (5 nm band pass, 90° polarizer) (SPEX FluoroMax 3 spectrofluorometer) and corrected for vesicle scattering. For lipid partitioning experiments, LUVs composed of a mix of 1:1 molar ratios of POPC (1-palmitoyl-2-oleoyl-sn-glycero-3-phosphocholine) and POPG (1-palmitoyl-2-oleoyl-sn-glycero-3-[phospho-rac-(1-glycerol)]) were added to a solution of toxin (~2 µM final concentration), maintained at 25°C with continuous stirring in a total volume of 2 ml. Mole-fraction partitioning coefficients ($K_x$) were calculated by measuring the fluorescence intensity (F) at 320 nm and normalizing to the zero lipid fluorescence intensity ($F_0$) (*Ladokhin et al., 2000*; *Milescu et al., 2007*). $K_x$ was calculated based on the best fits of the following equation to the data: $F/F_0(L) = 1 + (F/F_0^{max} - 1)K_x[L]/([W] + K_x[L])$, where $F/F_0(L)$ is the change in fluorescence intensity for a given concentration of lipid, $F/F_0^{max}$ is the maximum fluorescence increase at high lipid concentrations, [L] is the average available lipid concentration (60% of total lipid concentration) and [W] is the molar concentration of water (55.3 M).

Protection of tryptophan fluorescence from acrylamide quenching was examined in the absence and presence of lipids (toxin: lipid = 1:500) by titration with increasing concentrations of acrylamide (*Milescu et al., 2007*). The Stern–Volmer quenching constant ($K_{SV}$) was calculated based on the best fits of the following equation to the data: $F_0/F = 1 + K_{SV}[Q]$, where $F_0$ and $F$ are fluorescence of the toxin in the absence and presence of acrylamide, and [Q] is the concentration of acrylamide. Dequenching of tryptophan fluorescence by addition of lipids to a pre-quenched solution of toxin and acrylamide is a sensitive and reliable method to determine the strength of lipid interactions for molecules which do not exhibit large spectroscopic changes in response to lipid binding (*Posokhov et al., 2007*). A solution containing ~ 2 µM toxin and 0.3 M acrylamide was stirred continuously with addition of increasing concentrations of LUVs composed of 1:1 mixture of POPC and POPG. Fluorescence spectra (averaging three spectra) were recorded between 300 and 450 nm (5 nm band pass, 0° polarizer) using an excitation wavelength of 280 nm (5 nm band pass, 90° polarizer) (SPEX FluoroMax 3 spectrofluorometer) and corrected for vesicle scattering. For calculating mole–fraction partitioning coefficients ($K_{dx}$), the maximal fluorescence intensity (F at $\lambda_{max}$) was measured and normalized to the zero lipid fluorescence intensity ($F_0$). $K_{dx}$ was calculated based on the best fits of the following equation to the data: $F/F_0(L) = 1 + (F/F_0^{max} - 1)K_{dx}[L]/([W] + K_{dx}[L])$, where $F/F_0(L)$ is the change in maximum fluorescence intensity for a given concentration of lipid, $F/F_0^{max}$ is the maximum fluorescence increase at highest lipid concentration, [L] is the average available lipid concentration (60% of total lipid concentration) and [W] is the molar concentration of water (55.3 M).

For depth-dependent fluorescence quenching experiments with brominated lipids, LUVs contained a 1:1 molar ratio of POPG and POPC 6,7-, 9,10-, or 11,12-1-palmitoyl-2-stearoyl(dibromo)-sn-glycero-3-phosphocholine (diBr; Avanti Polar Lipids, Alabaster, AL). Fluorescence spectra were recorded between 300 and 450 nm using an excitation wavelength of 280 nm, corrected for vesicle scattering and normalized to the zero lipid fluorescence intensity.

## Construction of toxin-Kv channel models

Structural modeling of GxTx-1E—Kv1.2-Kv2.1 chimera complexes was performed using ROSETTA (*Gray et al., 2003*; *Rohl et al., 2004*; *Bradley et al., 2005*; *Wang et al., 2007*; *Raman et al., 2009*; *Tyka et al., 2011*; *Conway et al., 2014*). For each Kv2.1-ASIC1a chimera expressed, residues of S3b segment from the crystal structure of a Kv1.2-Kv2.1 channel (PDB 2R9R) (*Long et al., 2007*) were aligned in register with residues of ASIC1a helix-5 from the ASIC1a:PcTx1 crystal structure (PDB 4FZ0) (*Baconguis and Gouaux, 2012*). Then a GxTx-1E NMR structure (PDB 2WH9) (*Lee et al., 2010*) was aligned with PcTx1 structure to minimize RMSD between cystine α-carbons of the two toxins. Each GxTx-1E—Kv1.2-Kv2.1 chimera complex model was the relaxed using ROSETTA to identify the lowest energy binding models. ROSETTA binding energies at the toxin—channel interface ($\Delta\Delta G$) were calculated from in silico alanine scans as described previously (*Kortemme and Baker, 2002*; *Kortemme et al., 2004*).

## Acknowledgements

We thank members of the Swartz lab for helpful discussions. This work was supported by the Intramural Research Program of the NINDS, NIH to KJS, by NIH T32HL086350 for DCT, by 1U01NS090581 to JTS and VY-Y, and by the Basic Science Research Program through the National Research Foundation of Korea, funded by the Ministry of Education, Science and Technology (2013R1A1A2009798) to JIK.

## Additional information

### Funding

| Funder | Grant reference | Author |
|---|---|---|
| National Institute of Neurological Disorders and Stroke (NINDS) | Intramural Research Program, ZIA NS002945-18 | Kenton J Swartz |
| National Institutes of Health (NIH) | T32HL086350 | Drew C Tilley |
| Ministry of Education, Science and Technology | Basic Science Research program through the National Research Foundation of Korea, 2013R1A1A2009798 | Jae Il Kim |
| National Institute of Neurological Disorders and Stroke (NINDS) | 1U01NS090581 | Jon T Sack, Vladimir Yarov-Yarovoy |

The funders had no role in study design, data collection and interpretation, or the decision to submit the work for publication.

### Author contributions

KG, MZ, CB, KJS, Conception and design, Acquisition of data, Analysis and interpretation of data, Drafting or revising the article, Contributed unpublished essential data or reagents; MM, DK, Conception and design, Acquisition of data, Analysis and interpretation of data; DCT, VY-Y, Conception and design, Acquisition of data, Analysis and interpretation of data, Drafting or revising the article; JTS, Conception and design, Analysis and interpretation of data, Drafting or revising the article; JIK, Conception and design, Drafting or revising the article, Contributed unpublished essential data or reagents

### Ethics

Animal experimentation: This study was performed in strict accordance with the recommendations in the Guide for the Care and Use of Laboratory Animals of the National Institutes of Health. All of the animals were handled according to approved institutional animal care and use committee (IACUC) protocols (#1253-15) of the National Institute of Neurological Disorders and Stroke.

## Additional files

### Major datasets

The following previously published datasets were used:

| Author(s) | Year | Dataset title | Dataset ID and/or URL | Database, license, and accessibility information |
| --- | --- | --- | --- | --- |
| Baconguis I, Gouaux E | 2012 | Crystal structure of acid-sensing ion channel in complex with psalmotoxin 1 at low pH | http://www.rcsb.org/pdb/explore/explore.do?structureId=4FZ0 | Publicly available at RCSB Protein Data Bank (Accession No. 4FZ0). |
| Long SB, Tao X, Campbell EB, MacKinnon R | 2007 | Shaker family voltage dependent potassium channel (kv1.2-kv2.1 paddle chimera channel) in association with beta subunit | http://www.rcsb.org/pdb/explore/explore.do?structureId=2R9R | Publicly available at RCSB Protein Data Bank (Accession No. 2R9R). |
| Saez NJ, Mobli M, Bieri M, Chassagnon IR, Malde AK, Gamsjaeger R, Mark AE, Gooley PR, Rash LD, King GF | 2011 | High-resolution solution structure of the ASIC1a blocker PcTX1 | http://www.rcsb.org/pdb/explore/explore.do?structureId=2KNI | Publicly available at RCSB Protein Data Bank (Accession No. 2KNI). |
| Lee S, Milescu M, Jung HH, Lee JY, Bae CH, Lee CW, Kim HH, Swartz KJ, Kim JI | 2010 | Solution structure of GxTX-1E | http://www.rcsb.org/pdb/explore/explore.do?structureId=2WH9 | Publicly available at RCSB Protein Data Bank (Accession No. 2WH9). |

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
