## [Decision Letter]

Thank you for sending your work entitled “Tarantula toxins use common surfaces for interacting with Kv and ASIC ion channels” for consideration at *eLife*. Your article has been favorably evaluated by John Kuriyan (Senior editor), a Reviewing editor, and two reviewers.

The following individuals responsible for the peer review of your submission have agreed to reveal their identity: Rick Aldrich (Reviewing editor); Chris Miller (peer reviewer). A further reviewer remains anonymous.

The Reviewing editor and the reviewers discussed their comments before we reached this decision, and the Reviewing editor has assembled the following comments to help you prepare a revised submission.

This manuscript by Gupta et al. compares membrane interaction properties and channel binding surfaces of the ASIC1-targeting toxin PcTx1 with GxTx-1E, a toxin targeting potassium channels. The authors conclude that voltage-sensor toxins (GxTx-1E) partition deeper into the bilayer than PcTx1 which does not target voltage-sensors. Moreover, the authors suggest that both toxins have evolved a concave surface for interacting with alpha-helices whether this happens in aqueous or lipid environments. All investigators involved in this study are experts in their respective fields, which is reflected in the meticulous execution of the reported experiments. Altogether, this manuscript does a good job at convincing the reader of the author's conclusions. The manuscript is well written and well-referenced. The strongest and most important contribution to the field is that tarantula toxins of similar structure do not necessarily have similar membrane interacting properties which may reflect on their channel selectivity/potency properties.

The following points need to be addressed in a revised version:

1) For a non-expert, the effect of PcTx1 on membrane partitioning using Trp fluorescence seems non-significant (Figure 2). Can the authors elaborate on why they think this is significant in the corresponding manuscript section? Maybe it is worth showing fluorescence emission spectra for PcTx1 at higher lipid concentrations? We do understand that ensuing experiments attempt to verify the effect shown in Figure 2 but readers may get confused here. Also, do the authors expect to see a difference in membrane partitioning when adding in other lipids such as sphingomyelin and cholesterol to make the vesicles more mammalian-like (e.g. Cao et al., Neuron. 2013 77(4):667-79)? Finally, can the authors refer to (or perform) a control experiment under similar conditions with a peptide that is know not to partition in the membrane at all (if this exists)?

2) The Kdx value of PcTx1 obtained using the acrylamide experiment seems very different than that obtained using blue-shifts in Trp fuorescence whereas that of GxTx-1E is in reasonable agreement with the value obtained in Figure 2. Is there an explanation for this discrepancy? Is it possible that the two toxins present different surfaces to lipid vesicles which may alter the Trp that is presented to the lipid membrane?

3) Have the authors tried to insert longer S3b-S4 segments of ASIC1a into Kv2.1 to make the channel sensitive to PcTx1? From their previous publications on sodium channel constructs, it seems that transferring regions between channels is not always straightforward and may require some ingenuity (e.g. alignment gaps are frequently observed).

4) If GxTx-1E can partition deeply into the membrane to interact with the voltage sensor in Kv2.1 and PcTx1 cannot, but both toxins share a mechanism to latch onto alpha-helices, is it unreasonable to expect an effect of GxTx-1E on ASIC channels? Have the authors tested the effect of (maybe higher concentrations) GxTx-1E on ASICs?

---

## [Author Response]

*1) For a non-expert, the effect of PcTx1 on membrane partitioning using Trp fluorescence seems non-significant (*Figure 2*). Can the authors elaborate on why they think this is significant in the corresponding manuscript section? Maybe it is worth showing fluorescence emission spectra for PcTx1 at higher lipid concentrations? We do understand that ensuing experiments attempt to verify the effect shown in*
Figure 2
*but readers may get confused here. Also, do the authors expect to see a difference in membrane partitioning when adding in other lipids such as sphingomyelin and cholesterol to make the vesicles more mammalian-like (e.g. Cao et al., Neuron. 2013 77(4):667-79)? Finally, can the authors refer to (or perform) a control experiment under similar conditions with a peptide that is know not to partition in the membrane at all (if this exists)*?

The blue shifts we observed in Trp Fluorescence for PcTx1 as liposomes are added to aqueous solutions of the toxin are indeed very small. However, we think these blue shifts are significant and suggestive of an interaction between the toxin and membranes because they are reproducible and titrate with the amount of lipid added to the cuvette, as shown in the data plotted in Figure 2. Having said that, we also thought that it was crucial to verify this initial suggestion of an interaction between PcTx1 and membranes by asking whether membranes protect the toxin from quenching by acrylamide, as shown in Figure 2 and Figure 3. We have revised this section of the manuscript to clarify our thinking and we have removed any quantitative analysis of these data due to the very small blue shifts observed with PcTx1 (see Results, third paragraph).

We have not undertaken experiments with PcTx1 and more mammalian-like vesicles. It would be interesting, but we do not have a clear expectation of what we would find. In the case of the voltage-sensor toxins SGTx1, we can detect an interaction between the toxin and *Xenopus* oocyte membranes (containing both cholesterol and sphingomyelin), but these are relatively weak and difficult to quantify (see [51] in the reference list). The weak nature of the interaction of SGTx1 with more mammalian-like membranes led us to conclude that membrane partitioning will not lead to large increases in the toxin concentration within the membrane, but would certainly be sufficient for the toxin to access its receptor (the voltage-sensing domains) within the membrane.

It is an interesting point about whether any small protein toxin would interact with membranes with properties similar to what we observed with PcTx1. In response to the final point above, we examined the interaction between membrane vesicles and charybdotoxin, a pore-blocking scorpion toxin that contains one Trp residue surrounded by basic residues, which we suspected might not interact with membranes. However, we observed measurable (but small) blue shifts in Trp fluorescence, suggesting that charybdotoxin does interact with membranes. Moreover, a previous study investigated the interaction of radiolabeled charybdotoxin with membrane vesicles and the toxin was found to interact with membranes through an electrostatic mechanism ([6], Biophysical J. 73, 1717-1727.) We now cite this paper and discuss the relevance to our findings with PcTx1 on the second paragraph of the Discussion.

*2) The Kdx value of PcTx1 obtained using the acrylamide experiment seems very different than that obtained using blue-shifts in Trp fuorescence whereas that of GxTx-1E is in reasonable agreement with the value obtained in*
Figure 2*. Is there an explanation for this discrepancy? Is it possible that the two toxins present different surfaces to lipid vesicles which may alter the Trp that is presented to the lipid membrane*?

We attribute this difference to the very small blue shifts observed with PcTx1, as discussed above. We have now removed the fits to the titration experiment in Figure 1 to avoid confusion. The blue shifts in Trp fluorescence for PcTx1 can only be taken as a qualitative measure of a membrane interaction (which itself required validation), and not as a quantitative measure of the strength of membrane interaction.

Although we have not determined the orientation of either GxTx-1E or PcTx1 with membranes, we think that it is likely that they present different surfaces to membranes as their amphipathic character in their surfaces overlap, but are clearly distinct. This can be appreciated by comparing Figure 1, and appreciating that GxTx-1E in panel F has been rotated by 90° relative to the backbone superposed orientations shown in panels C, D and H.

*3) Have the authors tried to insert longer S3b-S4 segments of ASIC1a into Kv2.1 to make the channel sensitive to PcTx1? From their previous publications on sodium channel constructs, it seems that transferring regions between channels is not always straightforward and may require some ingenuity (e.g. alignment gaps are frequently observed)*.

We have not tried to make larger chimeras (beyond transferring helix 5) because most of the buried surface of PcTx1 when bound to ASIC involves interactions with residues in helix 5. PcTx1 does contain an arginine finger motif composed of R26, R27 and R28, that inserts into a crevice situated at the subunit interface in ASIC housing the protonated residues responsible for proton activation (4). However, the residues of the channel interacting with the Arg finger motif are located in regions discontinuous with helix 5, and thus would not be amenable to making chimeras. We have added several sentences to the Results section to explain these structural issues to the reader.

*4) If GxTx-1E can partition deeply into the membrane to interact with the voltage sensor in Kv2.1 and PcTx1 cannot, but both toxins share a mechanism to latch onto alpha-helices, is it unreasonable to expect an effect of GxTx-1E on ASIC channels? Have the authors tested the effect of (maybe higher concentrations) GxTx-1E on ASICs*?

It is possible that GxTx-1E might be able to bind to ASICs at high concentrations, but it would not be able to activate the channel because it lacks the Arg finger of PcTx1 that is required for activation of the channel by protons.